# Evaluation of Feature Extraction and Classification for Lower Limb Motion Based on sEMG Signal

**DOI:** 10.3390/e22080852

**Published:** 2020-07-31

**Authors:** Pengjie Qin, Xin Shi

**Affiliations:** Institute of Automation, Chongqing University, Chongqing 400044, China

**Keywords:** surface electromyography (sEMG), feature extraction, classification, lower limb motion

## Abstract

The real-time and accuracy of motion classification plays an essential role for the elderly or frail people in daily activities. This study aims to determine the optimal feature extraction and classification method for the activities of daily living (ADL). In the experiment, we collected surface electromyography (sEMG) signals from thigh semitendinosus, lateral thigh muscle, and calf gastrocnemius of the lower limbs to classify horizontal walking, crossing obstacles, standing up, going down the stairs, and going up the stairs. Firstly, we analyzed 11 feature extraction methods, including time domain, frequency domain, time-frequency domain, and entropy. Additionally, a feature evaluation method was proposed, and the separability of 11 feature extraction algorithms was calculated. Then, combined with 11 feature algorithms, the classification accuracy and time of 55 classification methods were calculated. The results showed that the Gaussian Kernel Linear Discriminant Analysis (GK-LDA) with WAMP had the highest classification accuracy rate (96%), and the calculation time was below 80 ms. In this paper, the quantitative comparative analysis of feature extraction and classification methods was a benefit to the application for the wearable sEMG sensor system in ADL.

## 1. Introduction

Due to the aging of the population, an increasing amount of elderly or weak people need help in daily life [1,2,3]. With the development of wireless networks and wearable sensor technology, a wearable sensor can sense the human body’s biological signal and classify the movement mode or body posture [4,5,6]. The auxiliary equipment based on the surface electromyography (sEMG) sensing systems, such as a rehabilitation robot and booster robot, can help the elderly or weak people to lead a better life [7,8,9].

The sEMG sensor measures the potential generated by muscle activity. The sEMG signals are generated in the range of 30 to 150 ms before human motion [10,11]. Therefore, the prediction of human motion can be realized by feature extraction and classification technology.

The sEMG is recorded from the surface of the human skeletal muscle by the surface electromyographic electrode, which contains much essential information related to limb movement. The key problem of these studies is to extract effective features from signals according to different motions [12]. The feature extraction methods of the sEMG signal mainly include time domain, frequency domain, and time-frequency domain. Among them, time-domain analysis is the most commonly used method such as integrated sEMG (IsEMG), mean absolute value (MAV), simple squared integration (SSI), root mean squared (RMS), wavelength (WL), zero-crossing (ZC), and Willison amplitude (WAMP) [13]. FEIYUN XIAO et al. used root mean square, waveform length, the absolute standard deviation of difference, integrated sEMG signal (IsEMG), and sEMG low-pass filtered (50 Hz) signal (LPFEMG) features to quickly and accurately estimate joint motion [14]. Osama dorgham et al. used time-domain features (such as MAV, RMS, VAR, and STD) to estimate muscle strength under different loads [15]. Shengli Zhou et al. used the frequency domain analysis method of median frequency (MDF) and peak frequency (PKF) to extract features, and combined with the Gaussian model, the accuracy of motion classification reached 89.5% [16]. Erdem Yavuz et al. extracted sEMG signal features by calculating Mel-Frequency Cepstral Coefficients (MFCCs) for basic motion classification [17]. The time-frequency domain analysis method can extract a large amount of information from the sEMG signal, among which the wavelet transform (WT) feature extraction method is the research hotspot in recent decades. Turker tuncer et al. used the iterative feature extraction method of discrete wavelet and tested the human muscle force with an sEMG data set, and the classification accuracy was 92.96% [18]. C. Sravani et al. used the flexible analytic wavelet transform (FAWT) to decompose the sEMG signal into eight sub-bands and extracted useful features, and the average accuracy of human motion classification was 91.5% [19]. Xugang Xi et al. used wavelet transforms to decompose sEMG signals into 32 scale signals and extracted sEMG features through coherent analysis to classify six movements of lower limbs, and the average classification rate was 93.45% [20]. Haotian she et al. used the time-frequency analysis method of the Stockwell transform (S-transform) and principal component analysis (PCA) to reduce the feature vector’s dimension and improve classifier operation speed, the average classification accuracy was 93.62% [21]. Hongfeng Chen et al. used the feature extraction method of the convolutional neural network (CNN) to improve the accuracy of human motion classification [22]. However, sEMG is a non-stationary, complex, and nonlinear signal. The entropy measurement method can reflect the sEMG signal’s complexity, which helps extract the effective features of the sEMG signal [23,24,25]. Shangchun Liao et al. used the method of integrating the entropy feature and wavelength feature of samples to realize the classification of human upper limb motion. Without expensive hardware support, the calculation was small, and the accuracy was 91.05% [26].

Another critical step of human motion classification is the selection of classification technology. Based on the above feature extraction methods, researchers mainly used support a vector machine (SVM), decision tree (DT), random forest (RF), nearest neighbor (KNN), and naive Bayes (NB) to classify human motion [27,28,29,30,31,32]. Rohit Gupta et al. used a time-domain analysis method to classify the movement of the lower limbs, and concluded that the linear discriminant analysis (LDA) classifier had the highest accuracy, and for different feature subsets, the classification accuracy was between 89% and 99% [33]. AI Qingsong et al. extracted the wavelet coefficients of sEMG, used linear discriminant analysis (LDA), and a support vector machine (SVM) based on the Gaussian kernel function to classify the lower limb motion accuracy higher than 95% [34]. In recent years, the neural network has been widely used in human complex motion classification because of its powerful nonlinear fitting function [35,36,37,38,39]. Chen Yang et al. extracted RMS, WC, and PE features of sEMG signals using the backpropagation neural network, generalized regression neural network, and least square support vector regression (LS-SVR) to predict the knee angle; the root mean square error was less than 7.7°, which can be used in a rehabilitation robot [40]. Lina Tong et al. used the Butterworth filtering method to extract sEMG features, and proposed a joint angle estimation method for real-time sEMG signals based on the backpropagation (BP) neural network and autoregressive (AR) model; the delay of this algorithm was 10 ~ 15 ms (PC), and the average angle RMS error was 4.27° [41].

Thanks to the surface electromyography, the surface of human skeletal muscle is recorded through surface electromyography electrodes and contains many feature information related to limb motion. By analyzing these features, we can distinguish daily human activities of the lower limb. Meanwhile, for the system with good performance, when selecting sEMG signal features, the features with maximum class separability, high recognition accuracy, and minimum computational complexity should be selected to ensure the high stability of auxiliary equipment. As far as the authors knew, there was almost no quantitative performance comparison of the feature extraction and classification methods for lower limb sEMG in daily human activities. Therefore, the purpose of this study was to determine the optimal sEMG features and classification methods.

The rest of this paper’s structure is as follows: Section 2 outlines the daily activities and data acquisition of human lower limbs. Section 3 analyzes the feature extraction method and classification method of the sEMG signal and proposed a feature evaluation method. The experimental results are given in Section 4. The discussion and conclusions are given in Section 5 and Section 6.

## 2. Data Acquisition

We chose the five most common activities of lower limbs in our daily life: horizontal walking (HW), crossing obstacles (CO), standing up (SU), going down the stairs (DS), and going up the stairs (GU). By analyzing the kinematics and biological characteristics of human lower limb muscles, the inner side of the gastrocnemius muscle (MG) is helpful for walking and running; the lateral femoral muscle (VL) and semitendinosus (ST) have the function of flexing the knee joint and stretching the hip joint. Therefore, we selected the above three muscles as the source of myoelectric signal acquisition, as shown in Figure 1.

We used an sEMG acquisition system developed and manufactured by Biometrics UK, as shown in Figure 2. The sampling frequency was 2000 Hz, and the amplifier’s input impedance was higher than 10,000,000 M Ohms. The skin did not require the conductive gel for processing to obtain better signal quality. The experimental computing platform processor was Intel (R) Core (TM) i7-9750H CPU@ 2.60GHz, the memory was 16 G, and the data analysis software was MATLAB2015b.

## 3. Algorithm Description

### 3.1. Feature Extraction

For each motion, we collected sEMG signals of 2 s (4000 sample points) and analyzed 11 common sEMG feature extraction methods, as shown in Table 1.

(1) Root mean square (RMS)

The root mean squared value (RMS) revealed the amount of strength yielded by a muscle.
(1)$$RMS=\sqrt{\frac{1}{N}\sum_{n=1}^{N}x_{n}{}^{2}}$$
where $x_{n}$ was the sample data, N was the sample length, which was 4000.

(2) Variance (VAR)

The VAR measured the power of the myoelectric signal.
(2)$$VAR=\frac{1}{N-1}\sum_{i=1}^{N}x_{i}{}^{2}$$
where $x_{i}$ was the sample data and N was sample length, which was 4000.

(3) Wilson Amplitude (WAMP)

Through the Willison amplitude, the number of times that two adjacent samples overcame a threshold was counted, reducing artifacts produced by noise.
(3)$$WAMP=\frac{1}{N}\sum_{n=1}^{N}f(\left|x_{n}\right|),\;f(x)=\left\{ \begin{array}{l} 1,x\geq th\\ 0,otherwise \end{array}\right.$$
where $x_{n}$ was the sample data and N was the sample length, which was 4000.

(4) Zero-Crossing (ZC)

The zero-crossing feature was to count the events produced by muscular activity.
(4)$$ZC=\sum_{i=1}^{N-1}u(-x_{i}x_{i+1})$$
where $x_{i}$ was the sample data and N was the sample length, which was 4000.

(5) Mean of absolute value (MAV)

The mean of absolute value was a reflection of muscle contraction levels:(5)$$MAV=\frac{1}{N}\sum_{i=1}^{N}\left|x_{i}\right|$$
where $x_{i}$ was the sample data and N was the sample length, which was 4000.

(6) Waveform length (WL)

The WL represented the amplitude, duration, and frequency of the signal.
(6)$$WL=\sum_{i=1}^{N}\left|x_{i}-x_{i-1}\right|$$
where $x_{i}$ was the sample data and N was the sample length, which was 4000.

(7) Integrated sEMG (IsEMG)

The IsEMG was related to the signal sequence firing point.
(7)$$IsEMG=\sum_{i=1}^{N}\left|x_{i}\right|$$
where $x_{i}$ was the sample data and N was the sample length, which was 4000.

(8) Simple squared integration (SSI)

The simple square integration function described the energy of the sEMG.
(8)$$SSI=\sum_{i=1}^{N}{\left|x_{i}\right|}^{2}$$
where $x_{i}$ was the sample data and N was the sample length, which was 4000.

(9) The Energy of Wavelet Packet Coefficient (EWP)

The EWP calculated the energy of the wavelet packet transform signal. It can process both high-frequency components and low-frequency components.

(10) The Energy of Wavelet Coefficient (EWC)

This feature computed the energy of the wavelet-transformed signal.
(9)$$EWC_{j}=\sqrt{\frac{1}{K}\sum_{k=1}^{K}W_{j,k}^{2}}$$
where $EWC_{j}$ was the coefficient of wavelet energy. The K was the number of the j-th layer decomposed coefficient. The $W_{j,k}$ was the k-th coefficient of the j-th layer decomposed coefficient.

(11) Fuzzy entropy (FE)

The FE can describe the complexity of the sEMG signal and reflect the possibility of the new information in the signal.
(10)$$FuzzyEn=\ln\Phi^{m}(r)-\ln\Phi^{m+1}(r)$$

### 3.2. Feature Separability

The Euclidean distance (ED) was used to measure the distance for sample features. The longer the distance, the greater the difference between sample features. The standard deviation (SD) was used to measure the dispersion for sample features. The smaller the standard deviation, the more stable the sample features.

We used the ratio between ED and SD that we called the RES index as a feature statistic measured metrics. The ED(m,n) was defined as
(11)$$ED(m,n)=\sqrt{{\left(m_{1}-n_{1}\right)}^{2}+{\left(m_{2}-n_{2}\right)}^{2}}$$
where m and n represented two of the three feature sets. The SD was defined as
(12)$$SD=\sqrt{\frac{\sum_{w=1}^{N_{W}}{\left(r_{w}-\sigma \right)}^{2}}{N_{W}}}$$
where $r_{w}$ represented the eigenvalue and $N_{W}$ represented the feature set size. The RES index was defined as
(13)$$RES(m,n)=\frac{ED(m,n)}{\bar{SD}}.$$

Additionally, we standardized the features then calculated the RES index. The normalization for features $F_{norm}$ was performed, and it was defined as
(14)$$F_{norm}=\frac{F+\min(F)}{\max(F+\min(F))}.$$

Obviously, as the RES index value increased, we could extract the best feature value.

### 3.3. Classification

We considered and listed the following five representative classification algorithms, as shown in Table 2.

(1) Multiple Kernel Relevance Vector Machine (MKRVM)

Different kernel functions correspond to feature spaces of different sEMG signals. The multi-kernel function is more capable of describing complex lower limb motion. Therefore, using the multiple kernel relevance vector machines (MKRVM) can improve the accuracy of lower limb motion classification.

(2) Random Forest (RF)

The random forest algorithm measures each feature’s contribution to the classification and ranks them according to the random forest algorithm’s evaluation criteria. In this way, we can understand important features in the feature set, which is very helpful for how to improve the feature classification.

(3) Backpropagation neural network (BPNN)

The BPNN is a network trained according to the error backpropagation algorithm. Because of its strong nonlinear fitting ability, it is widely used in human motion classification based on sEMG signals.

(4) Gaussian Kernel Linear Discriminant Analysis (GK-LDA)

The basic idea of the GK-LDA is to use Gaussian kernel functions to project high-dimensional vectors into low-dimensional vector spaces, causing the sample to have the largest inter-class distance and the smallest intra-class distance in the new subspace to improve the accuracy of the classification effect.

(5) Wavelet neural network (WNN)

The WNN is a local basis function network. The basis function has an adjustable resolution scale, causing the network to have a stronger nonlinear learning ability. The wavelet basis function has tight support, so the interaction between neurons is small, and the learning speed is faster.

## 4. Results

We used three sEMG sensors to classify five lower limb motion. The five motions were HW, CO, SU, DS, and GU, as shown in Figure 3.

Five healthy subjects, aged 23, 23, 25, 26, and 24, were selected to participate. The body fat rate was 17% ± 3%, and the height was 170 ± 5 cm. During the experiment, the subjects completed each action cycle in about 2 s. Therefore, no matter how long the action time was, we only needed to select the first 2 s of the sEMG data as the raw data for classification, which could recognize activities for a longer period of time. We collected sEMG signals from five movements of lower limbs per subject.

We recorded the changing trend in the three muscles’ sEMG signals in five movements, as shown in Figure 4.

There were significant signal changes in the three selected muscles during the lower limb movement. The sEMG signal will fluctuate only when the movement changes. Among the three myoelectric signals, the crossing obstacle was the most obvious. There were also similar sEMG signal maps for GU, HW, and CO, as DS and SU were hard to distinguish from the raw signal. The trend of each movement in the three channels was different, which helped to distinguish different movement, and verified the correctness of our muscle selection.

### 4.1. Feature Separability Results

To evaluate the performance of the feature extraction method, Figure 5 shows the scatterplot of 11 methods. Each feature extraction algorithm had three features after reduction. The scatter plots of the two features were extracted from five movements using 11 methods. Each movement used a specific color and ten sampling points. From Figure 5, we could see the feature separability of 11 feature extraction algorithms. Figure 5b,j,k had better performance.

The performance of 11 feature extraction algorithms was evaluated by the RES index defined by Formula (13). The results are shown in Table 3. The RES indexes of EWP were 15.1, 12.9, and 12.5, respectively. The RES indexes of WAMP were 12.0, 11.2, and 8.0, respectively. The RES indexes of FE were 13.4, 14.5, and 6.5, respectively. The EWP, WAMP, and FE had good separability. The RES indexes of EWC were 13.6, 12.9, and 4.2, respectively. The RES indexes of IsEMG were 12.3, 9.2, and 8.4, respectively. The EWC and IsEMG had poor performance.

### 4.2. Movements Classification Results

The feature data sets of five movements were input into five classifiers (Table 2). We collected 1500 sets of data from five subjects. All classifiers used 5-fold cross-validation. The data set was divided into five subsets. Among these subsets, one of the subsets was selected as the test data, and the remaining four subsets were used as the training data. Figure 6 shows the average classification accuracy of the five classifiers with 11 feature algorithms. Figure 6 shows that RF and GK-LDA classification results were excellent for all feature algorithm, and the variance of GK-LDA was the smallest.

We drew a box graph to describe 11 feature algorithms’ classification accuracy, as shown in Figure 7a. According to Figure 7a, the EWP, WL, FE, and WAMP accuracy was higher than other feature algorithms, and the dispersion degree of the EWP was the lowest. Additionally, we drew a box graph to describe the classification accuracy of 5 classification algorithms, as shown in Figure 7b. According to Figure 7b, the GK-LDA classification accuracy was higher than other algorithms, and the dispersion was the lowest. It was proved that the GK-LDA classifier with EWP, WL, FE, and WAMP features has better classification performance.

The hyperparameter adjustment had an important influence on the classifier results. In the MKRVM and GK-LDA classification process, the parameter value was −10 to 10, the step size was 0.5, and the best result was selected in all experiments. In the WNN and BPNN classification process, the enumeration method was used to determine the training times which were set to 5000, the learning rate was 0.01, the training error was 0.001, the number of hidden neurons was 6, and the best classification result was selected. In the RF classification process, the enumeration method was used to determine that the number of trees was 10, and the number of features was 2 to obtain the best classification results.

The calculation time and classification accuracy are shown in Table 4. The GK-LDA using the WAMP feature ranked first at 96%. The GK-LDA classifier’s accuracy rate with the features of EWC, EWP, IsEMG, WL, FE, and WAMP reached more than 90%, which were satisfied for the accuracy of the lower limb classification. The calculation time of the GK-LDA was below 80 ms. Except for BPNN, the average accuracy of classifiers with EWP, FE, and WAMP features was more than 90%. It was proved that the EWP, FE, WAMP, and GK-LDA have excellent performance.

Figure 8 is a comparison chart of the average classification accuracy rate and average calculation time of 11 features in five classifiers. The WAMP, combined with five classifiers, had a better classification effect in real-time and accuracy.

The sensitivity analysis of the algorithm could determine the error of the motion classification, which was helpful in enhancing the practicability and safety of the system. Hence, we selected 800 testing sets to verify the sensitivity of 55 models. Table 5, Table 6, Table 7, Table 8 and Table 9 show the sensitivity metrics of 11 combined algorithms. Except for the BPNN algorithm, other algorithms had high accuracy in CO. GU classification, the average classification rate of GK-LDA, and RF algorithms were higher than other classification algorithms. The accuracy of the GU classification in the GK-LDA classifier was higher than 90%. All classifiers had the lowest HW classification accuracy, but the GK-LDA classifier accuracy was higher than other classifiers, reaching more than 80%. The classification accuracy of each action with WAMP, FE, and EWP feature classifiers was higher than other combined algorithms, reaching more than 88%. It was proved that the GK-LDA classifier with EWP, FE, and WAMP has high reliability.

Furthermore, we selected a female subject with a height of 155 cm and an age of 25 years. Then, we collected 500 samples as the testing samples. Because of the difference between the new subjects and the five subjects, the accuracy of the new subjects was lower than the five subjects. However, the accuracy of the GK-LDA classifier with EWP, FE, and WAMP features was 93%. The average classification accuracy of the GK-LDA was 90%, higher than BPNN, WNN, RF, and MKRVM. Additionally, the variance of the GK-LDA was the smallest, as shown in Figure 9. The adaptability of the GK-LDA was proved.

## 5. Discussion

For systems with excellent performance, the sEMG signal feature should be selected with maximum class separability, high recognition accuracy, and minimum computational complexity, ensuring a misclassification rate in implementation. It is helpful for the wide application of the wearable system based on the sEMG signal. Additionally, in many studies, the lower limb motion classification accuracy was improved by increasing the number of electrodes. However, in the actual lower limb movement process, the sEMG sensor will be disturbed by noise, the more sensors will lead to the instability for classification accuracy, and the wearer will also feel uncomfortable. Furthermore, with the increase in the number of electrodes, more data dimensions must be processed, and the calculation will increase. Therefore, this paper collected three muscle sEMG signals in the lower limb, which was in accord with the typical application of the lower limb auxiliary equipment. We proposed a feature evaluation method to analyze the effectiveness of feature extraction and compared the quantitative performance of the sEMG feature extraction and classification methods in daily human activities. We aimed to determine the optimal feature extraction and classification method, and provided guidance for the design of the lower extremity motion classification system based on sEMG. The results showed that the EWP, FE, and WAMP have better performance in the feature separability. The main reason is that the above methods can accurately measure the complexity of the sEMG signal and extract its features from multiple scales. Among them, the WAMP has the best performance in real-time. If the computation time of the entropy and wavelet packet transform can be reduced, they will be the best choice. The accuracy of the GK-LDA with EWC, EWP, IsEMG, WL, and FE features was more than 90%, and the response time was less than 80 ms. After the box plot analysis, the GK-LDA with EWP, FE, WL, and WAMP features has good classification reliability. Through sensitivity analysis, the CO and DS classification accuracies were the highest, reaching 100%. Although the accuracy of the HW classification was the lowest, it was still over 80%. This could satisfy the needs of daily activities. There were some limitations to this study. It was necessary to use a wearable sEMG system to collect the lower limb sEMG data of the elderly or patients in the actual environment. It was unknown how effective the algorithm was in the elderly or real patients’ lower extremity activities. Additionally, sEMG is a bioelectrical signal recorded from the muscle surface by electrodes, but it is easily affected by electrode aging, sweat, and external electromagnetic interference. Therefore, the interference compensation of the sEMG signal will be considered in the next step. Furthermore, this paper only considered the classification effect of discrete actions and did not consider the classification effect of continuous action changes. Hence, in the next step, we need to determine the starting position of different activities in the sEMG signal and obtain different active sEMG signal regions.

## 6. Conclusions

To improve the accuracy and real-time classification, we analyzed a series of feature extraction and classification methods. Additionally, we proposed a feature evaluation method to analyze the effectiveness of feature extraction. The conclusion was that the WAMP, FE, and EWP feature extraction methods were highly separable, and the WAMP calculation time was shorter. The GK-LDA was the best method in the lower limb classification. GK-LDA and WAMP was the best combination in sEMG feature extraction and classification. The results were helpful to the development of a wearable sEMG system. It has important implications for other sEMG signal-based devices, such as clinical assistive devices, walking assist devices, and robotics or prosthetic devices. The next step is to apply the algorithm to real situations and combine the existing sEMG sensor with the physical signal sensor, such as an accelerometer or gyroscope sensor, to improve the classification accuracy. Meanwhile, the classification effect of different movements in the lower limbs’ continuous activities for the elderly or actual patients should be considered. In future applications, these algorithms can also be used to predict the risk of falling, only needing to collect and analyze the sEMG signal when falling.

## Figures and Tables

**Figure 1 entropy-22-00852-f001:**
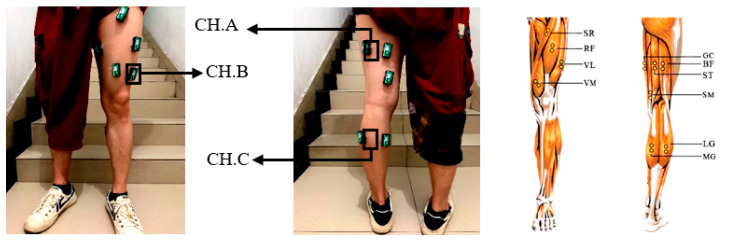
Surface electromyography (sEMG) signal sensor location. Channel A was located in the thigh semitendinosus, Channel B was located in the lateral thigh muscle, Channel C was located in the calf gastrocnemius.

**Figure 2 entropy-22-00852-f002:**
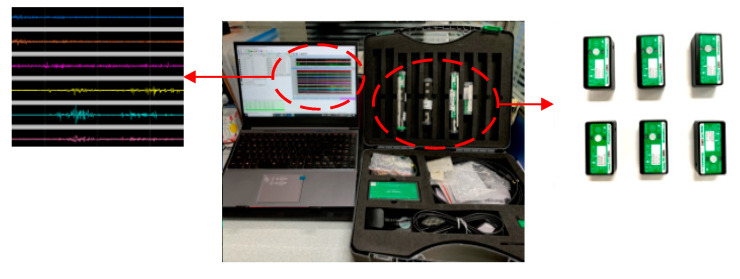
The biometrics wireless sEMG sensor system.

**Figure 3 entropy-22-00852-f003:**
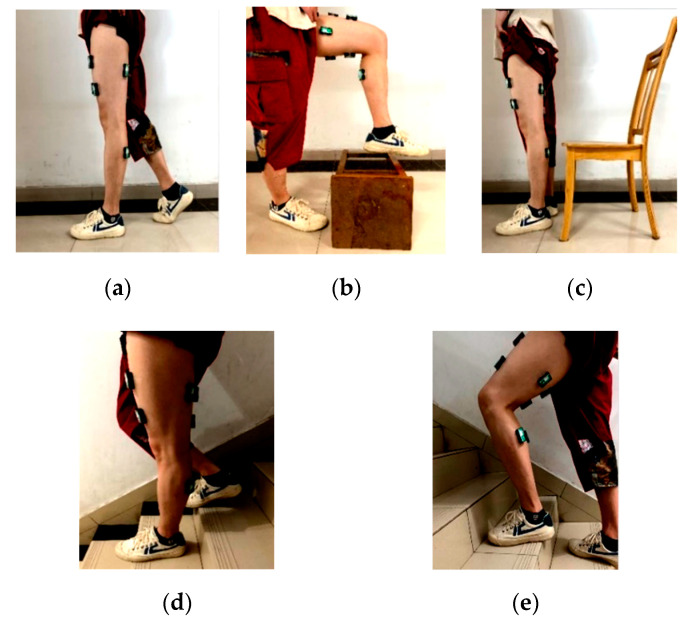
Five movements of lower limbs. (**a**) HW. (**b**) CO. (**c**) SU. (**d**) DS. (**e**) GU.

**Figure 4 entropy-22-00852-f004:**
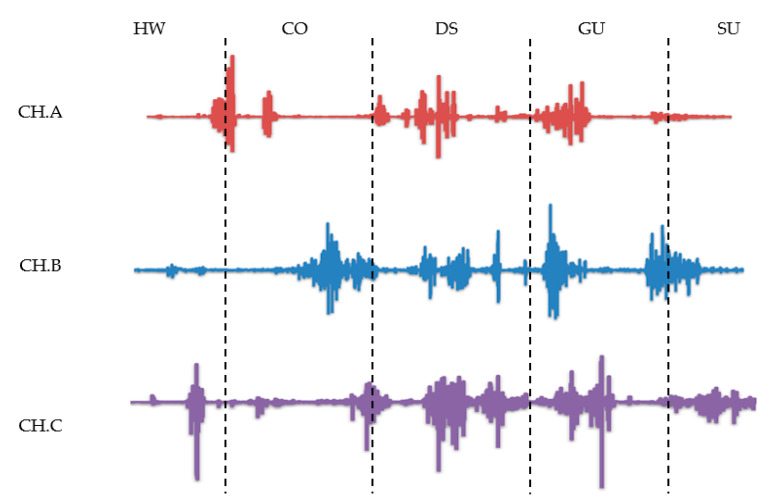
The raw sEMG signals of five movements.

**Figure 5 entropy-22-00852-f005:**
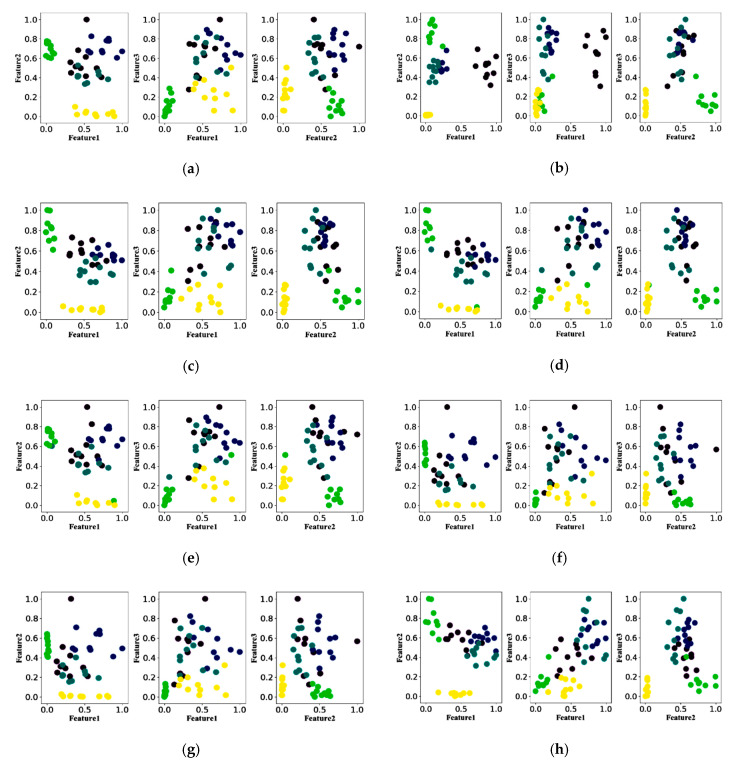
Separable scatterplot of eleven feature extraction algorithms. (**a**) Scatter plot for five different motion features extracted using the EWC. (**b**) Scatter plot for five different motion features extracted using the EWP. (**c**) Scatter plot for five different motion features extracted using the IsEMG. (**d**) Scatter plot for five different motion features extracted using the MAV. (**e**) Scatter plot for five different motion features extracted using the RMS. (**f**) Scatter plot for five different motion features extracted using the SSI. (**g**) Scatter plot for five different motion features extracted using the VAR. (**h**) Scatter plot for five different motion features extracted using the WL. (**i**) Scatter plot for five different motion features extracted using the ZC. (**j**) Scatter plot for five different motion features extracted using the FE. (**k**) Scatter plot for five different motion features extracted using the WAMP.

**Figure 6 entropy-22-00852-f006:**
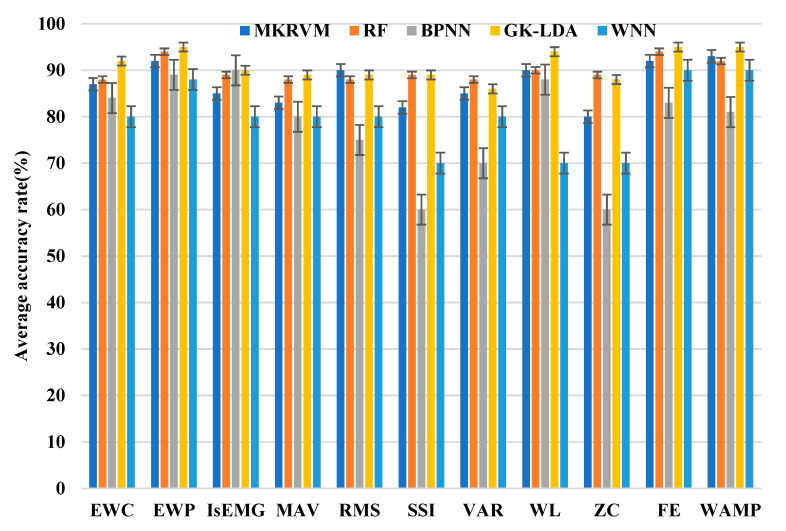
Average classification accuracy rates.

**Figure 7 entropy-22-00852-f007:**
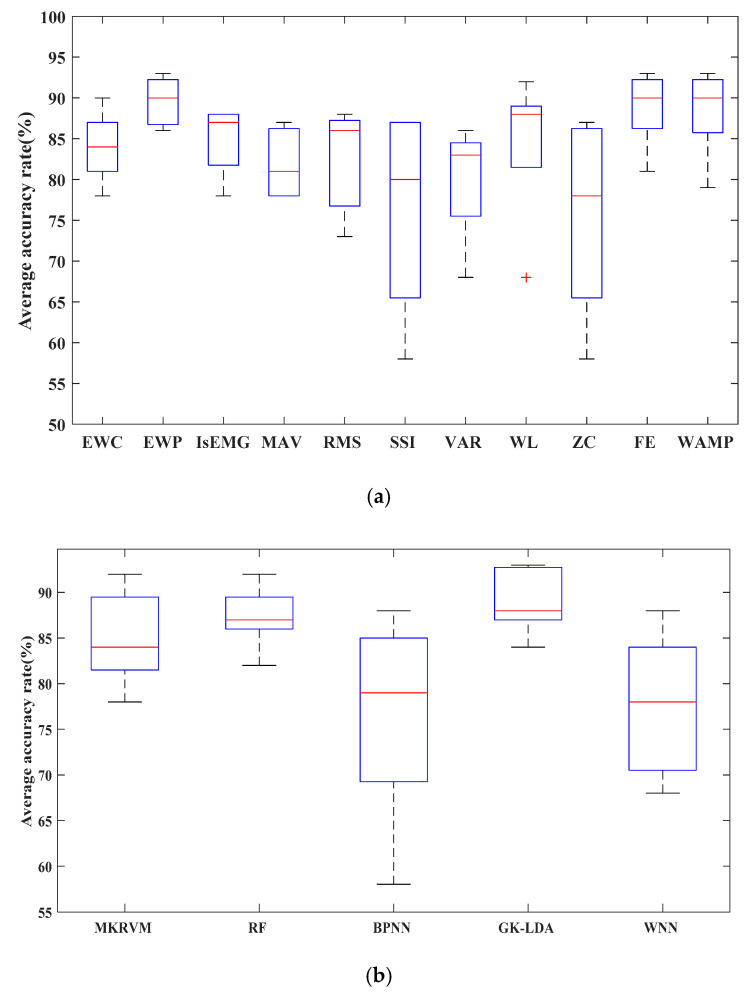
Box plot of the classification accuracy. (**a**) Box plot of 11 feature extraction algorithms. (**b**) Box plot of 5 classifier accuracies.

**Figure 8 entropy-22-00852-f008:**
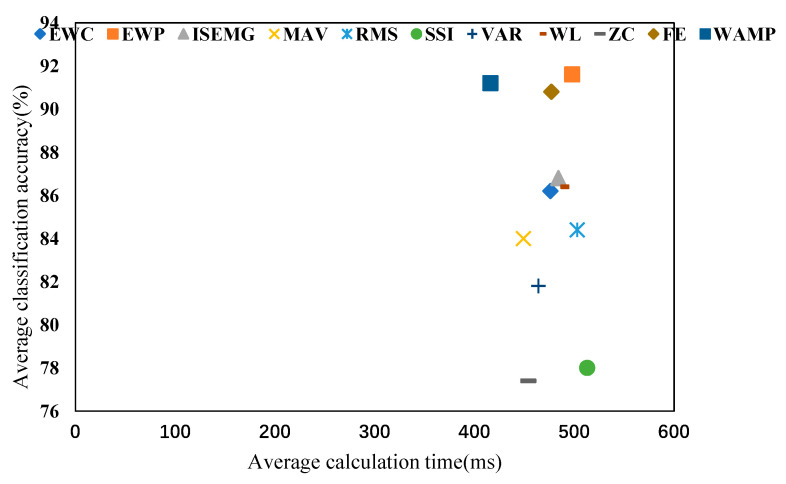
The average classification accuracy rate and the average calculation time across the classifiers.

**Figure 9 entropy-22-00852-f009:**
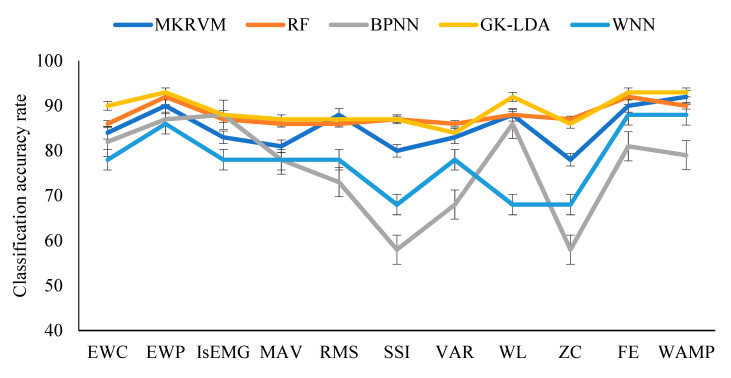
Average classification accuracy rates.

**Table 1 entropy-22-00852-t001:** Feature extraction method list.

ID	Extraction Feature	Abbreviation
1	Root mean square	RMS
2	Variance	VAR
3	Wilson Amplitude	WAMP
4	Zero-Crossing	ZC
5	Mean of absolute value	MAV
6	Waveform length	WL
7	Integrated sEMG	IsEMG
8	Simple squared integration	SSI
9	Energy of Wavelet Packet Coefficient	EWP
10	Energy of Wavelet Coefficient	EWC
11	Fuzzy entropy	FE

**Table 2 entropy-22-00852-t002:** Classification method list.

Classification Algorithm	Abbreviation
Multiple Kernel Relevance Vector Machine	MKRVM
Random Forest	RF
Back Propagation Neural Network	BPNN
Gaussian Kernel Linear Discriminant Analysis Linear Discriminate Analysis	GK-LDA
Wavelet Neural Network	WNN

**Table 3 entropy-22-00852-t003:** Statistical metrics of eleven feature extraction algorithms.

Algorithm	Feature Algorithm Metrics								
	Average Standard Deviation (SD)			Euclidean Distance (ED)			The Ratio between ED and SD (RES)		
	Feature 1	Feature 2	Feature 3	Feature 1	Feature 2	Feature 3	Feature 1	Feature 2	Feature 1
EWC	20.1	11.5	12.5	211.9	155.5	67.2	13.6	12.9	4.2
EWP	8.6	5.7	6.7	105.1	80.1	95.1	15.1	12.9	12.5
IsEMG	28,929.3	23,548.8	23,281.4	322,538.2	214,969.8	218,668.0	12.3	9.2	8.4
MAV	7.2	5.9	5.9	82.6	53.1	58.0	12.6	9.0	8.9
RMS	20.4	11.8	12.7	223.0	106.3	168.2	13.9	8.7	10.2
SSI	6,338,695	2,144,388	2,767,773	62,093,129.1	22,888,219.8	54,247,541.0	14.7	9.3	11.9
VAR	1582.6	536.4	689.7	15,481.3	5726.1	13,461.7	14.6	9.3	11.8
WL	2.7	1.6	1.6	29.2	14.2	20.8	13.4	8.9	9.7
ZC	81.9	59.2	32.1	702.8	617.6	480.7	10.1	13.7	8.6
FE	0.5	0.5	0.3	6.7	5.8	2.6	13.4	14.5	6.5
WAMP	6.1	10.9	5.8	102.6	93.7	47.7	12.0	11.2	8.0

**Table 4 entropy-22-00852-t004:** Classification rates and calculation time (ms, %). The double underscores represent the feature with the shortest computation time for each classifier. The underline represent the classifier with the shortest computation time for each feature. The shadow represents the most accurate feature of each classifier. The background shading represent the classifier with the highest accuracy for each feature.

Feature Extraction Method	Classification Algorithm									
	MKRVM		RF		BPNN		GK-LDA		WNN	
	Time	Accuracy	Time	Accuracy	Time	Accuracy	Time	Accuracy	Time	Accuracy
EWC	890	87	143	88	1027	84	69.8	**92**	255	80
EWP	969	92	146	94	1034	89	75.3	**95**	267	88
IsEMG	922	85	153	89	1044	**90**	59.4	**90**	246	80
MAV	934	83	146	88	856	80	59.8	**89**	254	80
RMS	998	**90**	149	88	1047	75	60.1	89	261	80
SSI	1058	82	149	**89**	1054	60	59.8	**89**	247	70
VAR	972	85	150	**88**	876	70	59.4	86	264	80
WL	1032	90	144	90	939	88	59.8	**94**	261	70
ZC	932	80	142	**89**	889	60	59.7	88	252	70
FE	895	92	188	94	1038	83	69.4	**95**	258	90
WAMP	863	93	145	92	764	81	59.9	**96**	253	90

**Table 5 entropy-22-00852-t005:** Sensitivity metrics of the MKRVM algorithm. The shadows represent the most accurate actions of each classifier. The bold numbers represent the classifier with the highest accuracy for each action.

Algorithm	Sensitivity (%)				
	HW	CO	SU	GU	DS
EWC_ MKRVM	85	89	86	91	84
EWP_ MKRVM	**89**	94	91	**96**	90
IsEMG_ MKRVM	83	87	84	89	82
MAV_ MKRVM	81	85	82	87	80
RMS_ MKRVM	88	92	89	94	87
SSI_ MKRVM	80	84	81	86	79
VAR_ MKRVM	80	89	85	88	83
WL_ MKRVM	85	94	90	93	88
ZC_ MKRVM	75	84	80	83	78
FE_ MKRVM	87	96	92	95	90
WAMP_ MKRVM	88	**98**	**93**	94	**92**

**Table 6 entropy-22-00852-t006:** Sensitivity metrics of the RF algorithm. The shadows represent the most accurate actions of each classifier. The bold numbers represent the classifier with the highest accuracy for each action.

Algorithm	Sensitivity (%)				
	HW	CO	SU	GU	DS
EWC_ RF	81	92	89	90	88
EWP_ RF	87	**98**	95	96	94
IsEMG_ RF	82	93	90	91	89
MAV_ RF	82	91	87	89	91
RMS_ RF	80	92	88	93	87
SSI_ RF	81	93	89	94	88
VAR_ RF	78	94	84	95	89
WL_ RF	80	96	86	97	91
ZC_ RF	83	91	89	92	90
FE_ RF	**90**	97	**96**	**99**	88
WAMP_ RF	88	**98**	92	**99**	83

**Table 7 entropy-22-00852-t007:** Sensitivity metrics of the BPNN algorithm. The shadows represent the most accurate actions of each classifier. The bold numbers represent the classifier with the highest accuracy for each action.

Algorithm	Sensitivity (%)				
	HW	CO	SU	GU	DS
EWC_ BPNN	82	85	84	86	83
EWP_ BPNN	**88**	**91**	**90**	**92**	**89**
IsEMG_ BPNN	87	90	89	91	88
MAV_ BPNN	78	81	80	82	79
RMS_ BPNN	73	76	75	77	74
SSI_ BPNN	58	61	60	62	59
VAR_ BPNN	68	71	70	72	69
WL_ BPNN	86	89	88	90	87
ZC_ BPNN	59	60	65	58	58
FE_ BPNN	81	84	83	85	82
WAMP_ BPNN	79	82	81	83	80

**Table 8 entropy-22-00852-t008:** Sensitivity metrics of the GK-LDA algorithm. The shadows represent the most accurate actions of each classifier. The bold numbers represent the classifier with the highest accuracy for each action.

Algorithm	Sensitivity (%)				
	HW	CO	SU	GU	DS
EWC_ GK-LDA	86	95	92	94	93
EWP_ GK-LDA	88	**99**	96	**97**	95
IsEMG_ GK-LDA	83	94	91	92	90
MAV_ GK-LDA	82	93	90	91	89
RMS_ GK-LDA	81	92	87	94	91
SSI_ GK-LDA	84	89	86	94	92
VAR_ GK-LDA	81	86	83	91	89
WL_ GK-LDA	**89**	94	91	**99**	**97**
ZC_ GK-LDA	83	88	85	93	91
FE_ GK-LDA	87	**100**	98	94	96
WAMP_ GK-LDA	85	**100**	**100**	97	93

**Table 9 entropy-22-00852-t009:** Sensitivity metrics of the WNN algorithm. The shadows represent the most accurate actions of each classifier. The bold numbers represent the classifier with the highest accuracy for each action.

Algorithm	Sensitivity (%)				
	HW	CO	SU	GU	DS
EWC_ WNN	75	82	79	83	81
EWP_ WNN	83	90	87	91	89
IsEMG_ WNN	77	85	76	82	80
MAV_ WNN	73	89	79	80	79
RMS_ WNN	80	82	77	81	80
SSI_ WNN	70	72	67	71	70
VAR_ WNN	79	83	78	81	79
WL_ WNN	69	73	68	70	70
ZC_ WNN	68	74	65	72	71
FE_ WNN	**89**	**93**	88	91	89
WAMP_ WNN	85	92	**89**	**93**	**91**

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
