# Peer review of "Evaluation of Feature Extraction and Classification for Lower Limb Motion Based on sEMG Signal"

_entropy, 2020, doi:10.3390/e22080852_

Round 1

Reviewer 1 Report

Congrats, very interesting paper.

All my comments are in the main text (PDF file).

I will add here just that one I consider most important.

INTRODUCTION

It is necessary a statement about why it is important to extract the lower limb sEMG in daily human activities. Better, why these results are useful?
If you answer this question on the introduction it will be fine.

Limitation

It's very important to make some statements about what is the EMG signal. In this case, it is important to mention the motor units. So, there are some changes in the motor unit patterns in the aging, and it can interfere in all algorithms.

Future

Just thinking in future applications, the algorithms could be used to predict risk of falls?

Review on discussion

Taking in mind the propose of the paper and the journal and considering the subjects are human people, the arguments in the discussion are disconnect from the human data. I mean, is not considered any physiological aspect. I have written some notes about physiological signals. Please, thinking carefully about these aspects and improve your discussion.
You can enlarge the audience, the paper will more useful for a large range of researches.

Reviewer 2 Report

This is a really interesting paper, good quality and clearly written. The article considers the most important EMG parameters including fuzzy entropy. The selection of parameters is performed correctly. The article provides references to the latest literature about sEMG feature extraction and classification methods. 

The choice of methodology is interesting and provides valuable information on the classification such type of signals, in the context of standard methods and more advanced ones, that taking into account the complex nature of kinesiological electromyography time series.

Some minor comments are presented below:

Introduction

lines 52-62

used instead of Used...

In general, in the whole Section, the Authors should investigate the correct use of capital letters.

The application of a specific type of entropy and its location against the other parameters are not fully explained. There are many different kinds of entropy estimators: sample, approximate, etc. Why the fuzzy entropy was the best choice for kinesiological sEMG?

In the discussion section, a broader discussion could be provided about the possible causes of differences in the accuracy of individual parameters and which of the parameters the Authors of this work see as potentially the most efficient in the kinesiological emg analysis. Is entropy the best candidate?
